# Unsupervised Feature Extraction by Time-Contrastive Learning and Nonlinear ICA

**Aapo Hyvärinen**[1,2] and **Hiroshi Morioka**[1]

[1] Department of Computer Science and HIIT
University of Helsinki, Finland

[2] Gatsby Computational Neuroscience Unit
University College London, UK

## Abstract

Nonlinear independent component analysis (ICA) provides an appealing framework for unsupervised feature learning, but the models proposed so far are not identifiable. Here, we first propose a new intuitive principle of unsupervised deep learning from time series which uses the nonstationary structure of the data. Our learning principle, time-contrastive learning (TCL), finds a representation which allows optimal discrimination of time segments (windows). Surprisingly, we show how TCL can be related to a nonlinear ICA model, when ICA is redefined to include temporal nonstationarities. In particular, we show that TCL combined with linear ICA estimates the nonlinear ICA model up to point-wise transformations of the sources, and this solution is unique — thus providing the first identifiability result for nonlinear ICA which is rigorous, constructive, as well as very general.

## 1 Introduction

Unsupervised nonlinear feature learning, or unsupervised representation learning, is one of the biggest challenges facing machine learning. Various approaches have been proposed, many of them in the deep learning framework. Some of the most popular methods are multi-layer belief nets and Restricted Boltzmann Machines [13] as well as autoencoders [14, 31, 21], which form the basis for the ladder networks [30]. While some success has been obtained, the general consensus is that the existing methods are lacking in scalability, theoretical justification, or both; more work is urgently needed to make machine learning applicable to big unlabeled data.

Better methods may be found by using the temporal structure in time series data. One approach which has shown a great promise recently is based on a set of methods variously called temporal coherence [17] or slow feature analysis [32]. The idea is to find features which change as slowly as possible, originally proposed in [6] for learning invariant features. Kernel-based methods [12, 26] and deep learning methods [23, 27, 9] have been developed to extend this principle to the general nonlinear case. However, it is not clear how one should optimally define the temporal stability criterion; these methods typically use heuristic criteria and are not based on generative models.

In fact, the most satisfactory solution for unsupervised deep learning would arguably be based on estimation of probabilistic generative models, because probabilistic theory often gives optimal objectives for learning. This has been possible in linear unsupervised learning, where sparse coding and independent component analysis (ICA) use independent, typically sparse, latent variables that generate the data via a linear mixing. Unfortunately, at least without temporal structure, the nonlinear ICA model is seriously unidentifiable [18], which means that the original sources cannot be found. In spite of years of research [20], no generally applicable identifiability conditions have been found. Nevertheless, practical algorithms have been proposed [29, 1, 5] with the hope that some kind of useful solution can still be found even for data with no temporal structure (that is, an i.i.d. sample).

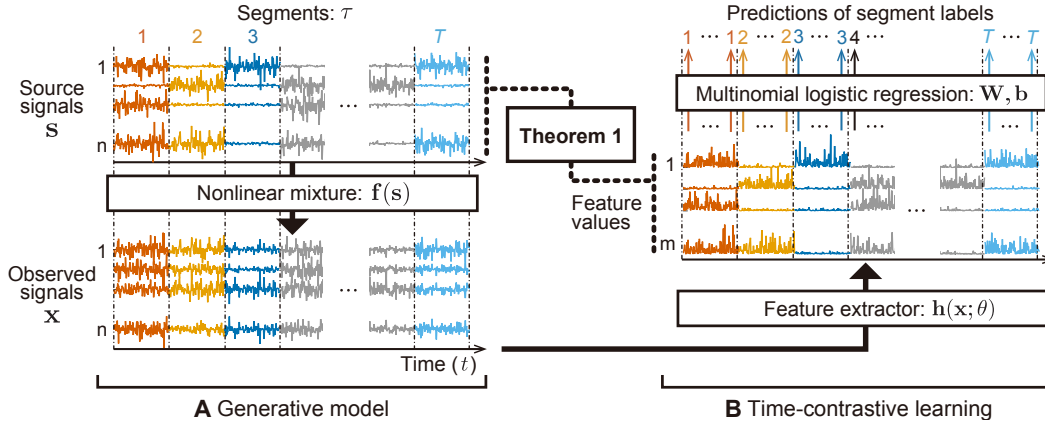

Figure 1: An illustration of how we combine a new generative nonlinear ICA model with the new learning principle called time-contrastive learning (TCL). *A)* The probabilistic generative model is a nonlinear version of ICA, where the observed signals are given by a nonlinear transformation of source signals, which are mutually independent, and have segment-wise nonstationarity. *B)* In TCL we train a feature extractor to be sensitive to the nonstationarity of the data by using a multinomial logistic regression which attempts to discriminate between the segments, labelling each data point with the segment label $1, \ldots, T$. The feature extractor and the logistic regression together can be implemented by a conventional multi-layer perceptron with back-propagation training.

Here, we combine a new heuristic principle for analysing temporal structure with a rigorous treatment of a nonlinear ICA model, leading to a new identifiability proof. The structure of our theory is illustrated in Figure 1.

First, we propose to learn features using the (temporal) nonstationarity of the data. The idea is that the learned features should enable discrimination between different time windows; in other words, we search for features that provide maximal information on which part of the time series a given data point comes from. This provides a new, intuitively appealing method for feature extraction, which we call time-contrastive learning (TCL).

Second, we formulate a generative model in which independent components have different distributions in different time windows, and we observe nonlinear mixtures of the components. While a special case of this principle, using nonstationary variances, has been very successfully used in linear ICA [22], our extension to the nonlinear case is completely new. Such nonstationarity of variances seems to be prominent in many kinds of data, for example EEG/MEG [2], natural video [17], and closely related to changes in volatility in financial time series; but we further generalize the nonstationarity to modulated exponential families.

Finally, we show that as a special case, TCL estimates the nonlinear part of the nonlinear ICA model, leaving only a simple linear mixing to be determined by linear ICA, and a final indeterminacy in terms of a component-wise nonlinearity similar to squaring. For modulated Gaussian sources, even the squaring can be removed and we have "full" identifiability. This gives the very first identifiability proof for a high-dimensional, nonlinear, ICA mixing model — together with a practical method for its estimation.

## 2 Time-contrastive learning

TCL is a method to train a feature extractor by using a multinomial logistic regression (MLR) classifier which aims to discriminate all segments (time windows) in a time series, given the segment indices as the labels of the data points. In more detail, TCL proceeds as follows:

1. Divide a multivariate time series $\mathbf{x}_t$ into segments, i.e. time windows, indexed by $\tau = 1, \ldots, T$. Any temporal segmentation method can be used, e.g. simple equal-sized bins.
2. Associate each data point with the corresponding segment index $\tau$ in which the data point is contained; i.e. the data points in the segment $\tau$ are all given the same segment label $\tau$.

3. Learn a feature extractor $\mathbf{h}(\mathbf{x}_t; \boldsymbol{\theta})$ together with an MLR with a linear regression function $\mathbf{w}_\tau^T \mathbf{h}(\mathbf{x}_t; \boldsymbol{\theta}) + b_\tau$ to classify all data points with the corresponding segment labels $\tau$ used as class labels $C_t$, as defined above. (For example, by ordinary deep learning with $\mathbf{h}(\mathbf{x}_t; \boldsymbol{\theta})$ being outputs in the last hidden layer and $\boldsymbol{\theta}$ being network weights.)

The purpose of the feature extractor is to extract a feature vector that enables the MLR to discriminate the segments. Therefore, it seems intuitively clear that the feature extractor needs to learn a useful representation of the temporal structure of the data, in particular the differences of the distributions across segments. Thus, we are effectively using a classification method (MLR) to accomplish unsupervised learning. Methods such as noise-contrastive estimation [11] and generative adversarial nets [8], see also [10], are similar in spirit, but clearly distinct from TCL which uses the *temporal* structure of the data by contrasting different time segments.

In practice, the feature extractor needs to be capable of approximating a general nonlinear relationship between the data points and the log-odds of the classes, and it must be easy to learn from data simultaneously with the MLR. To satisfy these requirements, we use here a multilayer perceptron (MLP) as the feature extractor. Essentially, we use ordinary MLP/MLR training according to very well-known neural network theory, with the last hidden layer working as the feature extractor. Note that the MLR is here only used as an instrument for training the feature extractor, and has no practical meaning after the training.

## 3  TCL as approximator of log-pdf ratios

We next show how the combination of the optimally discriminative feature extractor and MLR learns to model the nonstationary probability density functions (pdf's) of the data. The posterior over classes for one data point $\mathbf{x}_t$ in the multinomial logistic regression of TCL is given by well-known theory as

$$p(C_t = \tau | \mathbf{x}_t; \boldsymbol{\theta}, \mathbf{W}, \mathbf{b}) = \frac{\exp(\mathbf{w}_\tau^T \mathbf{h}(\mathbf{x}_t; \boldsymbol{\theta}) + b_\tau)}{1 + \sum_{j=2}^T \exp(\mathbf{w}_j^T \mathbf{h}(\mathbf{x}_t; \boldsymbol{\theta}) + b_j)} \tag{1}$$

where $C_t$ is a class label of the data at time $t$, $\mathbf{x}_t$ is the $n$-dimensional data point at time $t$, $\boldsymbol{\theta}$ is the parameter vector of the $m$-dimensional feature extractor (MLP) denoted by $\mathbf{h}$, $\mathbf{W} = [\mathbf{w}_1, \ldots, \mathbf{w}_T] \in \mathbb{R}^{m \times T}$, and $\mathbf{b} = [b_1, \ldots, b_T]^T$ are the weight and bias parameters of the MLR. We fixed the elements of $\mathbf{w}_1$ and $b_1$ to zero to avoid the well-known indeterminacy of the softmax function.

On the other hand, the true posteriors of the segment labels can be written, by the Bayes rule, as

$$p(C_t = \tau | \mathbf{x}_t) = \frac{p_\tau(\mathbf{x}_t) p(C_t = \tau)}{\sum_{j=1}^T p_j(\mathbf{x}_t) p(C_t = j)}, \tag{2}$$

where $p(C_t = \tau)$ is a prior distribution of the segment label $\tau$, and $p_\tau(\mathbf{x}_t) = p(\mathbf{x}_t | C_t = \tau)$.

Assume that the feature extractor has a universal approximation capacity (in the sense of well-known neural network theory), and that the amount of data is infinite, so that the MLR converges to the optimal classifier. Then, we will have equality between the model posterior Eq. (1) and the true posterior in Eq. (2) for all $\tau$. Well-known developments, intuitively based on equating the numerators in those equations and taking the pivot into account, lead to the relationship

$$\mathbf{w}_\tau^T \mathbf{h}(\mathbf{x}_t; \boldsymbol{\theta}) + b_\tau = \log p_\tau(\mathbf{x}_t) - \log p_1(\mathbf{x}_t) + \log \frac{p(C_t = \tau)}{p(C_t = 1)}, \tag{3}$$

where the last term on the right-hand side is zero if the segments have equal prior probability (i.e. equal length). In other words, what the feature extractor computes after TCL training (under optimal conditions) is the log-pdf of the data point in each segment (relative to that in the first segment which was chosen as pivot above). This gives a clear probabilistic interpretation of the intuitive principle of TCL, and will be used below to show its connection to nonlinear ICA.

## 4  Nonlinear nonstationary ICA model

In this section, seemingly unrelated to the preceding section, we define a probabilistic generative model; the connection will be explained in the next section. We assume, as typical in nonlinear ICA,

that the observed multivariate time series $\mathbf{x}_t$ is a smooth and invertible nonlinear mixture of a vector of source signals $\mathbf{s}_t = (s_1(t), \ldots, s_n(t))$; in other words:

$$\mathbf{x}_t = \mathbf{f}(\mathbf{s}_t). \tag{4}$$

The components $s_i(t)$ in $\mathbf{s}_t$ are assumed mutually independent over $i$ (but not over time $t$). The crucial question is how to define a suitable model for the sources, which is general enough while allowing strong identifiability results.

Here, we start with the fundamental assumption that the source signals $s_i(t)$ are *nonstationary*, and use such nonstationarity for source separation. For example, the variances (or similar scaling coefficients) could be changing as proposed earlier in the linear case [22, 24, 16]. We generalize that idea and propose a generative model for nonstationary sources based on the exponential family. Merely for mathematical convenience, we assume that the nonstationarity is much slower than the sampling rate, so the time series can be divided into segments in each of which the distribution is approximately constant (but the distribution is different in different segments). The log-pdf of the source signal with index $i$ in the segment $\tau$ is then defined as:

$$\log p_\tau(s_i) = q_{i,0}(s_i) + \sum_{v=1}^{V} \lambda_{i,v}(\tau) q_{i,v}(s_i) - \log Z(\lambda_{i,1}(\tau), \ldots, \lambda_{i,V}(\tau)) \tag{5}$$

where $q_{i,0}$ is a "stationary baseline" log-pdf of the source, and the $q_{i,v}, v \geq 1$ are nonlinear scalar functions defining the exponential family for source $i$; the index $t$ is dropped for simplicity. The essential point is that the parameters $\lambda_{i,v}(\tau)$ of the source $i$ depend on the segment index $\tau$, which creates nonstationarity. The normalization constant $Z$ disappears in all our proofs below.

A simple example would be obtained by setting $q_{i,0} = 0, V = 1$, i.e., using a single modulated function $q_{i,1}$ with $q_{i,1}(s_i) = -s_i^2/2$ which means that the variance of a Gaussian source is modulated, or $q_{i,1}(s_i) = -|s_i|$, a modulated Laplacian source. Another interesting option might be to use two nonlinearities similar to "rectified linear units" (ReLU) given by $q_{i,1}(s_i) = -\max(s_i, 0)$ and $q_{i,2}(s_i) = -\max(-s_i, 0)$ to model both changes in scale (variance) and location (mean). Yet another option is to use a Gaussian baseline $q_{i,0}(s_i) = -s_i^2/2$ with a nonquadratic function $q_{i,1}$.

Our definition thus generalizes the linear model [22, 24, 16] to the nonlinear case, as well as to very general modulated non-Gaussian densities by allowing $q_{i,v}$ to be non-quadratic, using more than one $q_{i,v}$ per source (i.e. we can have $V > 1$) as well as a non-stationary baseline. We emphasize that our principle of nonstationarity is clearly distinct from the principle of linear autocorrelations previously used in the nonlinear case [12, 26]. Note further that some authors prefer to use the term blind source separation (BSS) for generative models with temporal structure.

## 5   Solving nonlinear ICA by TCL

Now we consider the case where TCL as defined in Section 2 is applied on data generated by the nonlinear ICA model in Section 4. We refer again to Figure 1 which illustrates the total system. For simplicity, we consider the case $q_{i,0} = 0, V = 1$, i.e. the exponential family has a single modulated function $q_{i,1}$ per source, and this function is the same for all sources; we will discuss the general case separately below. The modulated function will be simply denoted by $q := q_{i,1}$ in the following.

First, we show that the nonlinear functions $q(s_i), i = 1, \ldots, n$, of the sources can be obtained as unknown linear transformations of the outputs of the feature extractor $h_i$ trained by the TCL:

**Theorem 1.** *Assume the following:*

  *A1. We observe data which is obtained by generating independent sources[1] according to (5), and mixing them as in (4) with a smooth invertible* $\mathbf{f}$. *For simplicity, we assume only a single function defining the exponential family, i.e.* $q_{i,0} = 0, V = 1$ *and* $q := q_{i,1}$ *as explained above.*

  *A2. We apply TCL on the data so that the dimension of the feature extractor* $\mathbf{h}$ *is equal to the dimension of the data vector* $\mathbf{x}_t$, *i.e.,* $m = n$.

*A3. The modulation parameter matrix $\mathbf{L}$ with elements $[\mathbf{L}]_{\tau,i} = \lambda_{i,1}(\tau) - \lambda_{i,1}(1), \tau = 1, \ldots, T; i = 1, \ldots, n$ has full column rank $n$. (Intuitively: the variances of the components are modulated sufficiently independently of each other. Note that many segments are actually allowed to have equal distributions since this matrix is typically very tall.)*

*Then, in the limit of infinite data, the outputs of the feature extractor are equal to $q(\mathbf{s}) = (q(s_1), q(s_2), \ldots, q(s_n))^T$ up to an invertible linear transformation. In other words,*

$$q(\mathbf{s}_t) = \mathbf{A}\mathbf{h}(\mathbf{x}_t; \boldsymbol{\theta}) + \mathbf{d} \tag{6}$$

*for some constant invertible matrix $\mathbf{A} \in \mathbb{R}^{n \times n}$ and a constant vector $\mathbf{d} \in \mathbb{R}^n$.*

*Sketch of proof*: (see Supplementary Material for full proof) The basic idea is that after convergence we must have equality between the model of the log-pdf in each segment given by TCL in Eq. (3) and that given by nonlinear ICA, obtained by summing the RHS of Eq. (5) over $i$:

$$\mathbf{w}_\tau^T \mathbf{h}(\mathbf{x}; \boldsymbol{\theta}) - k_1(\mathbf{x}) = \sum_{i=1}^{n} \lambda_{i,1}(\tau) q(s_i) - k_2(\tau) \tag{7}$$

where $k_1$ does not depend on $\tau$, and $k_2(\tau)$ does not depend on $\mathbf{x}$ or $\mathbf{s}$. We see that the functions $h_i(\mathbf{x})$ and $q(s_i)$ must span the same linear subspace. (TCL looks at differences of log-pdf's, introducing the baseline $k_1(\mathbf{x})$, but this does not actually change the subspace). This implies that the $q(s_i)$ must be equal to some invertible linear transformation of $\mathbf{h}(\mathbf{x}; \boldsymbol{\theta})$ and a constant bias term, which gives (6). $\qquad\square$

To further estimate the linear transformation $\mathbf{A}$ in (6), we can simply use linear ICA, under a further independence assumption regarding the generation of the $\lambda_{i,1}$:

**Corollary 1.** *Assume the $\lambda_{i,1}$ are randomly generated, independently for each $i$. The estimation (identification) of the $q(s_i)$ can then be performed by first performing TCL, and then linear ICA on the hidden representation $\mathbf{h}(\mathbf{x})$.*

*Proof:* We only need to combine the well-known identifiability proof of linear ICA [3] with Theorem 1, noting that the quantities $q(s_i)$ are now independent, and since $q$ has a strict upper bound (which is necessary for integrability), $q(s_i)$ must be non-Gaussian. $\qquad\square$

In general, TCL followed by linear ICA does not allow us to exactly recover the independent components because the function $q(\cdot)$ can hardly be invertible, typically being something like squaring or absolute values. However, for a specific class of $q$ including the modulated Gaussian family, we can prove a stricter form of identifiability. Slightly counterintuitively, we can recover the signs of the $s_i$, since we also know the corresponding $\mathbf{x}$ and the transformation is invertible:

**Corollary 2.** *Assume $q(s)$ is a strictly monotonic function of $|s|$. Then, we can further identify the original $s_i$, up to strictly monotonic transformations of each source.*

*Proof:* To make $p_\tau(s)$ integrable, necessarily $q(s) \to -\infty$ when $|s| \to \infty$, and $q(s)$ must have a finite maximum, which we can set to zero without restricting generality. For each fixed $i$, consider the manifold defined by $q(g_i(\mathbf{x})) = 0$. By invertibility of $\mathbf{g}$, this divides the space of $\mathbf{x}$ into two halves. In one half, define $\tilde{s}_i = q(s_i)$, and in the other, $\tilde{s}_i = -q(s_i)$. With such $\tilde{s}_i$, we have thus recovered the original sources, up to the strictly monotonic transformation $\tilde{s}_i = c \operatorname{sign}(s_i) q(s_i)$, where $c$ is either $+1$ or $-1$. (Note that in general, the $s_i$ are meaningfully defined only up to a strictly monotonic transformation, analogue to multiplication by an arbitrary constant in the linear case [3].) $\qquad\square$

**Summary of Theory**    What we have proven is that in the special case of a single $q(s)$ which is a monotonic function of $|s|$, our nonlinear ICA model is identifiable, up to inevitable component-wise monotonic transformations. We also provided a practical method for the estimation of the nonlinear transformations $q(s_i)$ for any general $q$, given by TCL followed by linear ICA. (The method provided for "inverting" $q$ in the proof of Corollary 2 may be very difficult to implement in practice.)

**Extensions**    First, allowing a stationary baseline $q_{i,0}$ does not change the Theorem at all, and a weaker form of Corollary 1 holds as well. Second, with many $q_{i,v}$ $(V > 1)$, the left-hand-side of (6) will have $Vn$ entries given by all the possible $q_{i,v}(s_i)$, and the dimension of the feature extractor must be equally increased; the condition of full rank on $\mathbf{L}$ is likewise more complicated. Corollary 1 must then consider an independent subspace model, but it can still be proven in the same way. (The details and the proof will be presented in a later paper.) Third, the case of combining ICA with dimension reduction is treated in Supplementary Material.

# 6 Simulation on artificial data

**Data generation** We created data from the nonlinear ICA model in Section 4, using the simplified case of the Theorem (a single function $q$) as follows. Nonstationary source signals ($n = 20$, segment length 512) were randomly generated by modulating Laplacian sources by $\lambda_{i,1}(\tau)$ randomly drawn so that the std's inside the segments have a uniform distribution in $[0, 1]$. As the nonlinear mixing function $\mathbf{f}(\mathbf{s})$, we used an MLP ("mixing-MLP"). In order to guarantee that the mixing-MLP is invertible, we used leaky ReLU's and the same number of units in all layers.

**TCL settings, training, and final linear ICA** As the feature extractor to be trained by TCL, we adopted an MLP ("feature-MLP"). The segmentation in TCL was the same as in the data generation, and the number of layers was the same in the mixing-MLP and the feature-MLP. Note that when $L = 1$, both the mixing-MLP and feature-MLP are a one layer model, and then the observed signals are simply linear mixtures of the source signals as in a linear ICA model. As in the Theorem, we set $m = n$. As the activation function in the hidden layers, we used a "maxout" unit, constructed by taking the maximum across $G = 2$ affine fully connected weight groups. However, the output layer has "absolute value" activation units exclusively. This is because the output of the feature-MLP (i.e., $\mathbf{h}(\mathbf{x}; \boldsymbol{\theta})$) should resemble $q(s)$, based on Theorem 1, and here we used the Laplacian distribution for generating the sources. The initial weights of each layer were randomly drawn from a uniform distribution for each layer, scaled as in [7]. To train the MLP, we used back-propagation with a momentum term. To avoid overfitting, we used $\ell_2$ regularization for the feature-MLP and MLR.

According to Corollary 1 above, after TCL we further applied linear ICA (FastICA, [15]) to the $\mathbf{h}(\mathbf{x}; \boldsymbol{\theta})$, and used its outputs as the final estimates of $q(s_i)$. To evaluate the performance of source recovery, we computed the mean correlation coefficients between the true $q(s_i)$ and their estimates. For comparison, we also applied a linear ICA method based on nonstationarity of variance (NSVICA) [16], a kernel-based nonlinear ICA method (kTDSEP) [12], and a denoising autoencoder (DAE) [31] to the observed data. We took absolute values of the estimated sources to make a fair comparison with TCL. In kTDSEP, we selected the 20 estimated components with the highest correlations with the source signals. We initialized the DAE by the stacked DAE scheme [31], and sigmoidal units were used in the hidden layers; we omitted the case $L > 3$ because of instability of training.

**Results** Figure 2a) shows that after training the feature-MLP by TCL, the MLR achieved higher classification accuracies than chance level, which implies that the feature-MLP was able to learn a representation of the data nonstationarity. (Here, chance level denotes the performance of the MLP with a randomly initialized feature-MLP.) We can see that the larger the number of layers is (which means that the nonlinearity in the mixing-MLP is stronger), the more difficult it is to train the feature-MLP and the MLR. The classification accuracy also goes down when the number of segments increases, since when there are more and more classes, some of them will inevitably have very similar distributions and are thus difficult to discriminate; this is why we computed the chance level as above.

Figure 2b) shows that the TCL method could reconstruct the $q(s_i)$ reasonably well even for the nonlinear mixture case ($L > 1$), while all other methods failed (NSVICA obviously performed very well in the linear case).The figure also shows that (1) the larger the number of segments (amount of data) is, the higher the performance of the TCL method is (i.e. the method seems to converge), and (2) again, more layers makes learning more difficult.

To summarize, this simulation confirms that TCL is able to estimate the nonlinear ICA model based on nonstationarity. Using more data increases performance, perhaps obviously, while making the mixing more nonlinear decreases performance.

# 7 Experiments on real brain imaging data

To evaluate the applicability of TCL to real data, we applied it on magnetoencephalography (MEG), i.e. measurements of the electrical activity in the human brain. In particular, we used data measured in a resting-state session, during which the subjects did not have any task nor were receiving any particular stimulation. In recent years, many studies have shown the existence of networks of brain activity in resting state, with MEG as well [2, 4]. Such networks mean that the data is nonstationary, and thus this data provides an excellent target for TCL.

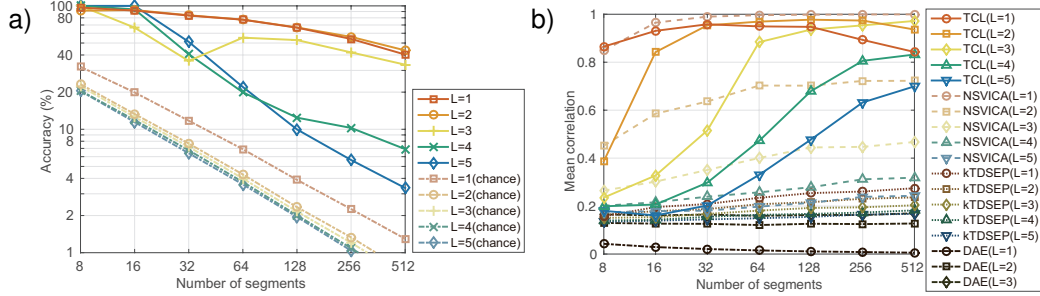

Figure 2: Simulation on artificial data. *a)* Mean classification accuracies of the MLR in TCL, as a function of the numbers of layers and segments. (Accuracies are on training data since it is not obvious how to define test data.) Note that chance levels (dotted lines) change as a function of the number of segments (see text). The MLR achieved higher accuracy than chance level. *b)* Mean absolute correlation coefficients between the true $q(s)$ and the features learned by TCL (solid line) and, for comparison: nonstationarity-based linear ICA (NSVICA, dashed line), kernel-based nonlinear ICA (kTDSEP, dotted line), and denoising autoencoder (DAE, dash-dot line). TCL has much higher correlations than DAE or kTDSEP, and in the nonlinear case ($L > 1$), higher than NSVICA.

**Data and preprocessing**  We used MEG data from an earlier neuroimaging study [25], graciously provided by P. Ramkumar. MEG signals were measured from nine healthy volunteers by a Vectorview helmet-shaped neuromagnetometer at a sampling rate of 600 Hz with 306 channels. The experiment consisted of two kinds of sessions, i.e., resting sessions (2 sessions of 10 min) and task sessions (2 sessions of 12 min). In the task sessions, the subjects were exposed to a sequence of 6–33 s blocks of auditory, visual and tactile stimuli, which were interleaved with 15 s rest periods. We exclusively used the resting-session data for the training of the network, and task-session data was only used in the evaluation. The modality of the sensory stimulation (incl. no stimulation, i.e. rest) provided a class label that we used in the evaluation, giving in total four classes. We preprocessed the MEG signals by Morlet filtering around the alpha frequency band.

**TCL settings**  We used segments of equal size, of length 12.5 s or 625 data points (downsampling to 50 Hz); the length was based on prior knowledge about the time-scale of resting-state networks. The number of layers took the values $L \in \{1, 2, 3, 4\}$, and the number of nodes of each hidden layer was a function of $L$ so that we always fixed the number of output layer nodes to 10, and increased gradually the number of nodes when going to earlier layer as $L = 1 : 10$, $L = 2 : 20 - 10$, $L = 3 : 40 - 20 - 10$, and $L = 4 : 80 - 40 - 20 - 10$. We used ReLU's in the middle layers, and adaptive units $\phi(x) = \max(x, ax)$ exclusively for the output layer, which is more flexible than the "absolute value" unit used in the Simulation above. To prevent overfitting, we applied dropout [28] to inputs, and batch normalization [19] to hidden layers. Since different subjects and sessions are likely to have uninteresting technical differences, we used a multi-task learning scheme, with a separate top-layer MLR classifier for each measurement session and subject, but a shared feature-MLP. (In fact, if we use the MLR to discriminate all segments of all sessions, it tends to mainly learn such inter-subject and inter-session differences.) Otherwise, all the settings were as in Section 6.

**Evaluation methods**  To evaluate the obtained features, we performed classification of the sensory stimulation categories (modalities) by applying feature extractors trained with (unlabeled) resting-session data to (labeled) task-session data. Classification was performed using a linear support vector machine (SVM) classifier trained on the stimulation modality labels, and its performance was evaluated by a session-average of session-wise one-block-out cross-validation (CV) accuracies. The hyperparameters of the SVM were determined by nested CV without using the test data. The average activities of the feature extractor during each block were used as feature vectors in the evaluation of TCL features. However, we used log-power activities for the other (baseline) methods because the average activities had much lower performance with those methods. We balanced the number of blocks between the four categories. We measured the CV accuracy 10 times by changing the initial values of the feature extractor training, and showed their average performance. We also visualized the spatial activity patterns obtained by TCL, using weighted-averaged sensor signals; i.e., the sensor signals are averaged while weighted by the activities of the feature extractor.

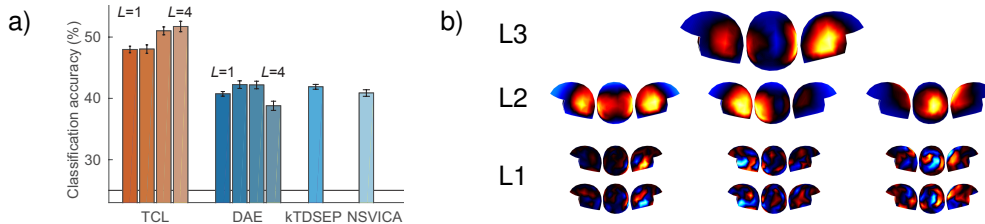

Figure 3: Real MEG data. *a)* Classification accuracies of linear SVMs newly trained with task-session data to predict stimulation labels in task-sessions, with feature extractors trained in advance with resting-session data. Error bars give standard errors of the mean across ten repetitions. For TCL and DAE, accuracies are given for different numbers of layers $L$. Horizontal line shows the chance level (25%). *b)* Example of spatial patterns of nonstationary components learned by TCL. Each small panel corresponds to one spatial pattern with the measurement helmet seen from three different angles (left, back, right); red/yellow is positive and blue is negative. *L3:* approximate total spatial pattern of one selected third-layer unit. *L2:* the patterns of the three second-layer units maximally contributing to this L3 unit. *L1:* for each L2 unit, the two most strongly contributing first-layer units.

**Results**  Figure 3a) shows the comparison of classification accuracies between the different methods, for different numbers of layers $L = \{1, 2, 3, 4\}$. The classification accuracies by the TCL method were consistently higher than those by the other (baseline) methods.[2] We can also see a superior performance of multi-layer networks ($L \geq 3$) compared with that of the linear case ($L = 1$), which indicates the importance of nonlinear demixing in the TCL method.

Figure 3b) shows an example of spatial patterns learned by the TCL method. For simplicity of visualization, we plotted spatial patterns for the three-layer model. We manually picked one out of the ten hidden nodes from the third layer, and plotted its weighted-averaged sensor signals (Figure 3b, L3). We also visualized the most strongly contributing second- and first-layer nodes. We see progressive pooling of L1 units to form left temporal, right temporal, and occipito-parietal patterns in L2, which are then all pooled together in the L3 resulting in a bilateral temporal pattern with negative contribution from the occipito-parietal region. Most of the spatial patterns in the third layer (not shown) are actually similar to those previously reported using functional magnetic resonance imaging (fMRI), and MEG [2, 4]. Interestingly, none of the hidden units seems to represent artefacts (i.e. non-brain signals), in contrast to ordinary linear ICA of EEG or MEG.

# 8  Conclusion

We proposed a new learning principle for unsupervised feature (representation) learning. It is based on analyzing nonstationarity in temporal data by discriminating between time segments. The ensuing "time-contrastive learning" is easy to implement since it only uses ordinary neural network training: a multi-layer perceptron with logistic regression. However, we showed that, surprisingly, it can estimate independent components in a nonlinear mixing model up to certain indeterminacies, assuming that the independent components are nonstationary in a suitable way. The indeterminacies include a linear mixing (which can be resolved by a further linear ICA step), and component-wise nonlinearities, such as squares or absolute values. TCL also avoids the computation of the gradient of the Jacobian, which is a major problem with maximum likelihood estimation [5].

Our developments also give by far the strongest identifiability proof of nonlinear ICA in the literature. The indeterminacies actually reduce to just inevitable monotonic component-wise transformations in the case of modulated Gaussian sources. Thus, our results pave the way for further developments in nonlinear ICA, which has so far seriously suffered from the lack of almost any identifiability theory, and provide a new *principled* approach to unsupervised deep learning.

Experiments on real MEG found neuroscientifically interesting networks. Other promising future application domains include video data, econometric data, and biomedical data such as EMG and ECG, in which nonstationary variances seem to play a major role.[3]

## Footnotes

[1]More precisely: the sources are generated independently given the $\lambda_{i,v}$. Depending on how the $\lambda_{i,v}$ are generated, there may or may not be marginal dependency between the $s_i$; see the Corollary 1 below.

[2]Note that classification using the final linear ICA is equivalent to using whitening since ICA only makes a further orthogonal rotation.

[3]This research was supported in part by JSPS KAKENHI 16J08502 and the Academy of Finland.

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
