[Supplementary Material · nipscameraready_suppl.pdf]

# Supplementary Material

*Unsupervised Feature Extraction by Time-Contrastive Learning and Nonlinear ICA*
*A. Hyvärinen and H. Morioka*
*NIPS 2016*

**Proof of Theorem**

We start by computing the log-pdf of a data point $\mathbf{x}$ in the segment $\tau$ under the nonlinear ICA model. Denote for simplicity $\lambda_{\tau,i} = \lambda_{i,1}(\tau)$. Using the probability transformation formula, the log-pdf is given by

$$\log p_\tau(\mathbf{x}) = \sum_{i=1}^{n} \lambda_{\tau,i} q(g_i(\mathbf{x})) + \log|\det \mathbf{J}\mathbf{g}(\mathbf{x})| - \log Z(\lambda_\tau), \tag{8}$$

where we drop the index $t$ from $\mathbf{x}$ for simplicity, $\mathbf{g}(\mathbf{x}) = (g_1(\mathbf{x}), \ldots, g_n(\mathbf{x}))^T$ is the inverse function of (the true) mixing function $\mathbf{f}$, and $\mathbf{J}$ denotes the Jacobian; thus, $s_i = g_i(\mathbf{x})$ by definition. By Assumption A1, this holds for the data for any $\tau$. Based on Assumptions A1 and A2, the optimal discrimination relation in Eq. (3) holds as well and is here given by

$$\log p_\tau(\mathbf{x}) = \sum_{i=1}^{n} w_{\tau,i} h_i(\mathbf{x}) + b_\tau + \log p_1(\mathbf{x}) - c_\tau, \tag{9}$$

where $w_{\tau,i}$ and $h_i(\mathbf{x})$ are the $i$th element of $\mathbf{w}_\tau$ and $\mathbf{h}(\mathbf{x})$, respectively, we drop $\boldsymbol{\theta}$ from $h_i$ for simplicity, and $c_\tau$ is the last term in (3).

Now, from Eq. (8) with $\tau = 1$, we have

$$\log p_1(\mathbf{x}) = \sum_{i=1}^{n} \lambda_{1,i} q(g_i(\mathbf{x})) + \log|\det \mathbf{J}\mathbf{g}(\mathbf{x})| - \log Z(\lambda_1). \tag{10}$$

Substituting Eq. (10) into Eq. (9), we have equivalently

$$\log p_\tau(\mathbf{x}) = \sum_{i=1}^{n} [w_{\tau,i} h_i(\mathbf{x}) + \lambda_{1,i} q(g_i(\mathbf{x}))] + \log|\det \mathbf{J}\mathbf{g}(\mathbf{x})| - \log Z(\lambda_1) + b_\tau - c_\tau. \tag{11}$$

Setting Eq. (11) and Eq. (8) to be equal for arbitrary $\tau$, we have:

$$\sum_{i=1}^{n} \tilde{\lambda}_{\tau,i} q(g_i(\mathbf{x})) = \sum_{i=1}^{n} w_{\tau,i} h_i(\mathbf{x}) + \beta_\tau, \tag{12}$$

where $\tilde{\lambda}_{\tau,i} = \lambda_{\tau,i} - \lambda_{1,i}$ and $\beta_\tau = \log Z(\lambda_\tau) - \log Z(\lambda_1) + b_\tau - c_\tau$. Remarkably, the log-determinants of the Jacobians cancel out and disappear here.

Collecting the equations in Eq. (12) for all the $T$ segments, and noting that by definition $\mathbf{s} = \mathbf{g}(\mathbf{x})$, we have a linear system with the "tall" matrix $\mathbf{L}$ in Assumption A3 on the left-hand side:

$$\mathbf{L}q(\mathbf{s}) = \mathbf{W}\mathbf{h}(\mathbf{x}) + \boldsymbol{\beta}, \tag{13}$$

where we collect the $\beta_\tau$ in the vector $\boldsymbol{\beta}$ and the $w_{\tau,i}$ in the matrix $\mathbf{W}$. Assumption A3 ($\mathbf{L}$ has full column rank) implies that its pseudoinverse fullfills $\mathbf{L}^+\mathbf{L} = \mathbf{I}$. We multiply the equation above from the left by such pseudoinverse and obtain

$$q(\mathbf{s}) = [\mathbf{L}^+\mathbf{W}]\mathbf{h}(\mathbf{x}) + \mathbf{L}^+\boldsymbol{\beta}. \tag{14}$$

Here, we see that the $q(s_i)$ are obtained as a linear transformation of the feature values $\mathbf{h}(\mathbf{x})$, plus an additional bias term $\mathbf{L}^+\boldsymbol{\beta}$, denoted by $\mathbf{d}$ in the Theorem. Furthermore, the matrix $\mathbf{L}^+\mathbf{W}$, denoted by $\mathbf{A}$ in the theorem, must be full rank (i.e. invertible), because if it were not, the functions $q(s_i)$ would be linearly dependent, which is impossible since they are each a function of a unique variable $s_i$. $\quad\square$

**Dimension reduction**

In practice we may want to set the feature extractor dimension $m$ to be smaller than $n$, to accomplish dimension reduction. It is in fact simple to modify the generative model and the theorem so that a dimension reduction similar to nonlinear PCA can be included, and performed by TCL. It is enough to assume that while in the nonlinear mixing (4) we have the same number of dimensions for both $\mathbf{x}$ and $\mathbf{s}$, in fact some of the components $s_i$ are stationary, i.e. for them, $\lambda_{\tau,i}$ do not depend on $\tau$. Then, the equations (12) regarding the log-pdfs of such stationary components will have zero on the left hand side because $\tilde{\lambda}_{\tau,i}$ will be zero for all $\tau$; the equality is trivially fullfilled by setting the corresponding $w_{\tau,i}$ to zero. Such stationary components thus have no effect on the learning procedure. The rest of the proof is not affected, and the nonstationary components $s_1(t), \ldots, s_m(t)$ will be identified as in the Theorem, using TCL.