[Reviews · NeurIPS 2016]

Reviewer 1

Summary

This paper proposes a new method for unsupervised learning of nonlinear features from non-stationary data. The features are tuned to be optimally discriminative between different time segments in the data. Interestingly, they show that this model can uniquely recover nonlinear independent components under certain conditions.

Qualitative Assessment

There has been a lot of progress taking the successes of supervised techniques and extending them to unsupervised domains by defining an interesting label to predict, and this paper continues this trend by highlighting the task of discriminating between time windows. Furthermore, I think the emphasis on the importance of identifiability for nonlinear ICA and showing how it is solved in this case is a timely contribution. However, these contributions were somewhat muted by other shortcomings of the approach which make me doubt the robustness and generality of the method. 1) Temporal "Artifacts", a systematic deficiency in the setup In the MRI results, the authors use a single MLP across all data but a separate MLR for each subject and session to avoid learning "artifactual differences". This is a significant source of prior knowledge to include in the experiments and makes the comparisons unfair. In particular the comment that "none of the hidden units seem to represent artefacts, in contrast to ICA" rings a bit hollow since it seems that the elimination of artifacts was really achieved through hand-picked choice in the model. The idea of trivial temporal artifacts that distinguish time periods is not, I think, an unlucky result of the MRI experimental setup either. I've noticed this occurring, e.g., in financial data where historical periods are trivially distinguished by the overall size of the market. If the method as described typically fails in the presence of these artifacts, I think its overall usefulness will be limited. The bright side is that putting in additional information about which types of temporal differences to learn really seems to help, so systematically exploring this effect could really strengthen the results. 2) Generality of the identifiability result unclear As the authors point out, independent components suffer from a bad identifiability problem, so I was most intrigued by Theorem 1. However, I immediately began to wonder about the generality of the result. Does it rely on the assumption that the discriminator has "universal approximation capacity"? To me this seems like a large and poorly defined assumption. Also in A1: "for simplicity we assume only a single function". The proof also assumes this, so it seems to be necessary for the proof? It seems like a large restriction, though it covers the non-stationary variance problem which seems to have been studied previously. Extension 2 suggests that the extension is straightforward, but if this is true, why not include this proof in the supplementary material? To me, it looks significantly harder. Minor point: - In some cases, there might be a natural way to split up time windows, or you can simply split them finely enough that the stationarity holds within each window. I guess you need enough samples in each window to estimate the parameters of a totally new model. Is it easy to estimate exactly how many this should be? edit: Thanks for the clarification about artifacts. I am still worried about it, but I think it is a general problem for unsupervised learning that could be addressed in future work. I.e., what if the best predictors of time slices are not good predictors for different tasks? Are there perhaps more systematic ways to force the method to learn interesting features (tying weights across subjects was certainly effective in this case)?

Confidence in this Review

2-Confident (read it all; understood it all reasonably well)


Reviewer 2

Summary

The paper presents a new technique for unsupervised feature extraction for time-series data and relates this approach to Nonlinear ICA. The approach operates by chopping time into intervals and then training a multinomial classifier to try to identify which interval a piece of data belongs to after they have been passed through a feature extractor (e.g. a deep network). They apply the approach on one set of artificial data that is generated from a NICA model and to one real-world dataset of brain imaging data. On both of these they show improvement over chosen baselines.

Qualitative Assessment

The manuscript presents an interesting new method for doing unsupervised feature extraction from time-series data, and demonstrates several theoretical results that tie the approach to NICA. The idea of using the multinomial regression as a way of producing features seems powerful in its simplicity. The experimental results are well executed, but are somewhat limited, thus making it difficult to judge how much impact the approach might have in the long run. In the single real-world problem that the approach is applied to it does markedly better than the selected baselines. But the problem is not, as far as I know, standard. Ideally, the approach could be validated on an existing benchmark for which there exist strong baselines (are there no other time-series datasets that would fit the bill here?). Failing this, the manuscript should do a better job of arguing for the importance of a new dataset+setting, and the strength of the chosen baselines for this new setting. Another possible avenue would be to show that the technique resolves a previously unresolved issue with EMG data in particular, but presently it is proposed as a general method and it would be preferable to see it retain this. The paper was generally very well written and organized and the diagrams are nicely presented. Minor Issues: One of the worries I had was with respect to how heuristic the time segmentation seems to be. This seems like a very important aspect of the approach and though Figure 2 speaks to the issue I do worry a bit about how to choose this so that things work well in the case of real data that may be on quite different scales. What is a reasonable number for $T$, the number of segments? Is there any recipe for this other than to try everything? Figure 3(b) is perhaps difficult to appreciate for those not expert in MEG. Perhaps the paper real estate could be used to better illustrate other, more general aspects of the performance of the algorithm? Grammar/Spelling issues on lines: 310 -- "inevitable"

Confidence in this Review

2-Confident (read it all; understood it all reasonably well)


Reviewer 3

Summary

The authors proposed a nonlinear ICA method from nonstationary time series. Using the nonstationary structure of data, the authors proposed the first Identifiability result for nonlinear ICA.

Qualitative Assessment

This paper studies the nonlinear ICA problem from nonstationary time series. The authors give the first Identifiability results on the nonlinear ICA using the nonstationary structure of data. The proposed time-contrastive learning (TCL) method learns a feature extractor to discriminate different segments of the time series and proved its relation to the source distribution. Also, the authors proposed corresponding estimation methods and validate the effectiveness on both artificial and real data. The experimental results on both artificial and real data validate the theoretical results of the paper. The theoretical result given in Theorem 3 of the paper is novel and interesting. The extracted feature is equal to q(s) up to an invertible linear transformation, which makes the identifiability possible. The three assumptions are not very strong in practice. One practical issue is the segments of the time series. The authors assume that the nonstationarity is much slower than the sampling rate so that the distribution is the same in the same segment and is different in different segments. What if two or more than two segments share the same distribution? Will the theoretical results hold in this case? In the real data experiments, it is better to illustrate the results of ICA for comparison of the spatial patterns. The aim of the proposed method is recover the independent components, while the authors reported the classification accuracy of time series segments. Is the classification accuracy of segments related to the recover quality? The paper is well-structured and most of the paper is written in a well-understandable way. However, it is better to use “artifact” or “artefact” rather than use both.

Confidence in this Review

2-Confident (read it all; understood it all reasonably well)


Reviewer 4

Summary

A time series classifier is trained over temporal data in order to learn temporally relevant features from that data. These features may then be used in other applications. One such application, a kind of MEG classification, is demonstrated.

Qualitative Assessment

I did not understand most of this paper. It appears to be well written but I honestly cannot say whether or not the ideas are sound or rational. I am equally unable to assess whether the experimental results are interesting. I like the idea of learning features imbued with the power to perform well on an inherently useful but ultimately unrelated task than the desired one; this is thus a form of multi task learning. The connection to ICA, which takes up the majority of the paper, was not particularly interesting to me, nor did I understand the parts that I tried to read. This should not be taken as a commentary on the paper quality, but should reflect a misfit of theory papers and practically-minded readers. The experimental section was somewhat weak; it was unclear whether the use of these features represents a new state of the art or any substantial improvement or whether the baselines are artificially low.

Confidence in this Review

1-Less confident (might not have understood significant parts)


Reviewer 5

Summary

The authors propose a new tractable method for nonlinear ICA on sequences called Time Contrastive Learning: by breaking sequences into time-indexed segments and learning to predict this time index from the segment features, one learns in the process the underlying independent components up to an affine transform. The authors demonstrate the proposed algorithm on a simulated dataset and MEG data and show reasonable improvement.

Qualitative Assessment

The theoretical framework of Time Contrastive Learning is quite strong and the derived theoretical results that this multinomial logistic regression results in an estimation of nonlinear independent components under some assumptions is quite interesting and novel. However I am not able to understand intuitively the extend to which extent predicting those time indices can be done in time series, especially if those time series are periodic or stationary. The importance of the segmentation for TCL is also not well understood from reading the paper. I believe those two points can greatly improve the understanding of the paper. The paper contains only two experimental setups including one simulated dataset but both seem to have been well executed, giving more understanding of the model in the simulated dataset and demonstrating reasonable performance with the MEG data.

Confidence in this Review

2-Confident (read it all; understood it all reasonably well)